# Three/Four-Dimensional Printed PLA Nano/Microstructures: Crystallization Principles and Practical Applications

**DOI:** 10.3390/ijms241813691

**Published:** 2023-09-05

**Authors:** Yufeng Zhou, Jingbo Chen, Xuying Liu, Jianwei Xu

**Affiliations:** School of Materials Science & Engineering, Zhengzhou University, Zhengzhou 450001, China; yfzhou823@163.com (Y.Z.); chenjb@zzu.edu.cn (J.C.); liuxy@zzu.edu.cn (X.L.)

**Keywords:** PLA, 3D/4D printing, nano/microstructures, crystallization

## Abstract

Compared to traditional methods, three/four-dimensional (3D/4D) printing technologies allow rapid prototyping and mass customization, which are ideal for preparing nano/microstructures of soft polymer materials. Poly (lactic acid) (PLA) is a biopolymer material widely used in additive manufacturing (AM) because of its biocompatibility and biodegradability. Unfortunately, owing to its intrinsically poor nucleation ability, a PLA product is usually in an amorphous state after industrial processing, leading to some undesirable properties such as a barrier property and low thermal resistance. Crystallization mediation offers a most practical way to improve the properties of PLA products. Herein, we summarize and discuss 3D/4D printing technologies in the processing of PLA nano/microstructures, focusing on crystallization principles and practical applications including bio-inspired structures, flexible electronics and biomedical engineering mainly reported in the last five years. Moreover, the challenges and prospects of 3D/4D printing technologies in the fabrication of high-performance PLA materials nano/microstructures will also be discussed.

## 1. Introduction

Three-dimensional printing, also known as AM or rapid prototyping technology, has become a powerful and multi-purpose technology for fabricating complicated 3D structures from all kinds of materials, such as ceramics, metals and polymers [1,2]. In 3D printing, the desired 3D objects are processed layer-by-layer from a digital model by adding material [3]. Three-dimensional printing is regarded as a new approach to enhance precision with a prominent reduction in time and costs by comparison with traditional subtractive manufacturing [4,5]. In 2013, Tibbits first proposed the concept of “4D printing” and regarded time as the fourth dimension [6]. The main difference between 3D printing and 4D printing is the addition of smart design and responsive materials which leads to an optimization of 3D printed structures over time. In order to modify the structure and function of 3D printing after printing, the material must have unique characteristics to respond to external stimuli. This responsiveness can enable printing equipment to be activated with specialized functions after the manufacturing process. In principle, 4D printing supports greater design freedom, which can be applied to novel applications using adaptive materials and improve the ability to print products. In 1959, Richard Feynman delivered a well-known lecture titled “There is Plenty of Room at the Bottom”, inspiring a notable increase in research at the nano/micro scale [7]. So far, nano- and microstructures have been widely used in metamaterials, bionics and optical components [8,9,10,11,12,13,14,15]. Three/four-dimensional printing plays a crucial role in the fabrication of nano- and microstructures. In 2020, Liashenko et al. [16] showed the printing of 3D objects with submicrometer features through high-speed amplifiers controlled by software to obtain ultrafast jet deflection. In 2021, Jung et al. [17] reported a technique for the direct 3D printing of arrays of metal nanostructures with feature sizes and flexible geometry shrinking to hundreds of nanometers by using diverse materials. In 2023, Esleane et al. [18] reported that in nanostructured 3D bioprinting of PLA with bioglass-CNT scaffolds for osseous tissue graft manufacturing, nanostructures act as a nucleation site for the growth of PLA crystals during the printing, which improves the compressive modulus of scaffolds.

From the polymer family, PLA has been recognized as a fascinating material for 3D/4D printing due to its biocompatibility, biodegradation and non-toxic behavior. Based on the chirality of monomers, PLA is classified as PLLA, PDLA and PDLLA. PLA exhibits a glass-transition temperature (*T*_g_) of about 60 °C, resulting in a main drawback of poor thermal resistance. PLA exhibits various types of crystal modifications, including the *α* form, the *δ* (or *α*′) form, the *β* form, the *γ* form and the stereocomplex between PLLA and PDLA. The crystal structure will be introduced in detail in a later section. Crystallization is a physical method for improving thermal resistance. In addition, many other properties (barrier, mechanical, etc.) can also be improved upon crystallization [19,20]. In contrast to other semicrystalline polymers, the crystallization rate of PLA is relatively low owing to poor nucleation ability. This review summarizes and discusses the 3D/4D printed PLA nano/microstructures that have been mainly published in the past five years, and it elucidates the relationship between fabrication processes and crystallization principles. The prospects of 3D/4D printed PLA in emerging applications such as bio-inspired structures, flexible electronics and biomedical engineering are summarized. In addition, the crystallization mechanism, including nucleation and crystal growth, is also described. We hope that this review can provide new directions and ideas for 3D/4D printed PLA nano/microstructures in practical applications.

## 2. Crystallization Principles

Crystallization is the most practical method for improving the properties of polymer materials. Here, the crystallization principles are introduced, mainly including the nucleation and growth of crystals.

### 2.1. Nucleation of Crystals

#### 2.1.1. Nucleation Theory Concept

It is well known that a nucleus can be defined as the minimum amount of a new crystalline phase. In 1878, the classical crystal nucleation theory was proposed by Gibbs and considered a free energy barrier for crystal nucleation due to a higher interfacial free energy penalty than the volume free energy gain for a small new phase [21,22,23], expressed by Equation (1)
(1)$$\Delta G=-\Delta g\times \frac{4}{3}\pi r^{3}+\sigma \times 4\pi r^{2}$$
where Δ*g* is the volume free energy gain per volume, *r* is the sphere radius of new phases and *σ* is the interfacial free energy density. The critical nucleus size *r** can be derived from a maximum in the free energy curve, as given by
(2)$$r^{*}=\frac{2\sigma }{\Delta g}$$
and the critical free energy barrier Δ*G** as obtained by
(3)$$\Delta G^{*}=\frac{16\pi \sigma^{3}}{3\Delta g^{2}}$$

Here, the equilibrium melting point $T_{m}^{0}$ and the crystallization enthalpy and entropy are insensitive to the crystallization temperature $T_{c}^{}$; thus, Δ*g* can be defined as
(4)$$\Delta g=\Delta h-T_{c}\Delta s\approx \Delta h-T_{c}\frac{\Delta h}{T_{m}^{0}}=\Delta h\frac{T_{m}^{0}-T_{c}}{T_{m}^{0}}\propto \Delta T$$

#### 2.1.2. Homogeneous Nucleation and Heterogeneous Nucleation

Crystallization is described by the process of primary nucleation and crystal growth. Primary nucleation involves homogeneous nucleation and heterogeneous nucleation. Homogeneous nucleation can be defined as a spontaneous aggregation of polymer molecules forming a nucleus with a critical size to overcome the barrier of the primary nucleation [24]. Because of the chain-like connection, the shape of macromolecules is anisotropic, which favors parallel packing in the crystalline states [25,26,27]. In heterogeneous nucleation, polymer chains typically prefer to nucleate on pre-existing surfaces because this process requires a dramatically lower energy barrier. The heterogeneous nucleation normally occurs in bulk polymers as they usually include a large number of heterogeneities, such as impurities, additives, catalytic debris or dust [28].

#### 2.1.3. Self-Nucleation

When semicrystalline polymers are melted between the nominal melting temperature and the equilibrium melting temperature, surviving ordered structures within the melts dramatically influence the nucleation, crystallization kinetics and morphology. Such a phenomenon is frequently referred to as self-nucleation or memory effect [29,30]. Self-nucleation has high nucleation efficiency because of a good lattice matching between surviving seeds and the polymer itself, which can induce crystallization at low supercooling. Zhang et al. [31] studied the structural evolution of the isotactic polypropylene (iPP) crystals during the melting process and found that the lamellar crystals transform into cylindrical nanocrystals in the later stage of melting (Figure 1b). Müller and co-workers [32,33] have employed the self-nucleation technique to control the nucleation density predictably. Yan et al. [34,35] demonstrated that partial melting of the iPP fiber can induce the formation of *β*-iPP.

#### 2.1.4. Flow-Induced Nucleation

Concerning the crystallization process, flow can induce the formation of oriented crystallites, raise nucleation density and oriented nuclei, and then accelerate the kinetics by orders of magnitude. In addition, flow can also induce conformational ordering. When the length or concentration of the ordering segments exceeds a certain threshold, the ordered segments may spontaneously aggregate and organize, e.g., in an isotropic–nematic transition [36,37]. Consequently, the inherent multiscale features of a polymer chain result in a multistep ordering process and multiscale ordered structures for flow-induced nucleation (Figure 1c) [38].

### 2.2. Growth of Crystals

#### 2.2.1. General Description of Crystal Growth

Compared to small molecules, polymers encounter enormous difficulty in the process of crystallization (Figure 1d) [39]. A polymer invariably arrives as a “chain of blocks”, rather than attaching individual “blocks” independently. They can only be integrated into the crystal at a limited number of adjacent sites due to the connectivity of the blocks. Therefore, polymers form lamellar crystals bound by surfaces that contain chain folds [40,41]. The lamellar thickness (i.e., the length of the crystalline stems) and thus the degree of chain order increase with time or temperature for polymer chains integrating into a crystal.

**Figure 1 ijms-24-13691-f001:**
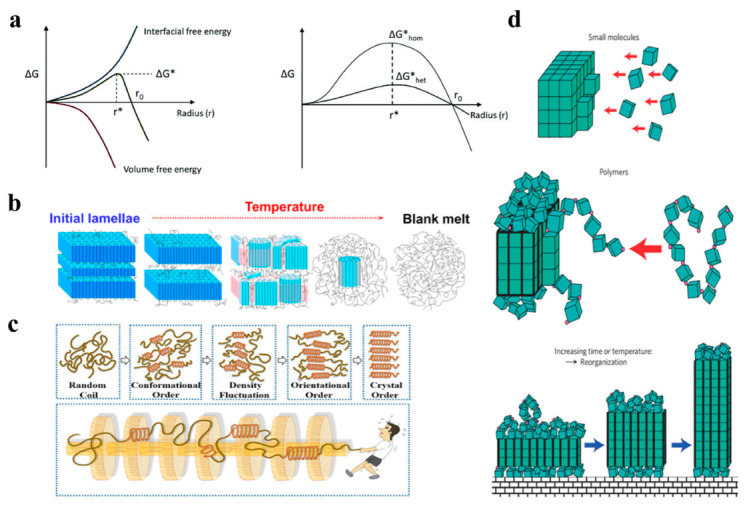
(**a**) Relationship between volume and interfacial free energy for the free energy barrier to form a nucleus (Reprinted with permission [42]. Copyright©2021, The Royal Society of Chemistry). (**b**) Schematic illustration of the melting process for polymer crystals (Reprinted with permission [31]. Copyright©2020, American Chemical Society). (**c**) Multistep ordering process and multiscale ordered structures of flow-induced nucleation for polymers (Reprinted with permission [38]. Copyright©2018, American Chemical Society). (**d**) Schematic presentation of crucial steps in polymer crystallization (Reprinted with permission [39]. Copyright©2009, Springer Nature).

#### 2.2.2. Solution Crystallization and Melt Crystallization

Most studies on solution crystallization have been carried out on polyethylene. In 1962, Holland and Lindenmeyer [43] observed the size of single crystals with time and found the linear growth rate on a logarithmic scale versus 1/*T*. The overall crystallization rates of 0.5 wt% polyethylene/*n*-hexadecane, *p*-xylene and decalin solutions were measured by Devoy et al. [44], and the results were similar to the data from bulk polymers. In addition, Miyata et al. [45] proposed that two molecular chains are located at the center and each corner of the unit cell for poly (L-lactide) solution-gown crystals. Hong et al. [46] discovered that PLLA samples from dilute solutions exhibited a higher crystallinity and faster crystallization rate.

Due to the constant thermodynamic driving force during the crystallization, the melt crystallization of a polymer appears to be a simpler process than solution crystallization. The growth rates of spherulite on polyethylene, isotactic polyethylene, poly (ethylene sebacate) and poly (ethylene oxide) from the melt and viscous concentrations were studied by Asaubekov [47] in 1972, and it was found that melt crystallization is faster than solution crystallization at the constant supercooling.

#### 2.2.3. Kinetics of Crystallization

The kinetics of crystallization is an important consideration in polymer processing as it significantly affects the final thermal and mechanical properties of materials. The Avrami equation (Equation (5)) is the most widely used method for analyzing the crystallization kinetics of polymers [48,49]. However, the limitations of this model have been found; for example, it is only applicable to primary crystallization. Some other models, including the simplified Hillier, Tobin, Malkin and Urbanovici–Segal models, many of which are extensions or adaptations of standard Avrami analysis, aimed at explaining the existence of secondary crystallization [50].
(5)$$X_{t}=1-e^{-kt^{n}{}^{}}$$
where *X*_t_ is the crystallinity, *k* is a kinetic rate constant and n is the Avrami exponent, which should be an integer between 1 and 4 for polymer crystallization and is associated with the nucleation mechanism (homogeneous vs. heterogeneous and simultaneous vs. sporadic).

To rapidly compare the crystallization rates of polymer materials, it is convenient to report the crystallization half-time (*t*_1/2_) defined as the time required to attain half of the final crystallinity (*X*_t_ = 0.5) [51]. The half-time is typically reported as a function of temperature, enabling the determination of the optimal temperature window.

### 2.3. Crystal Structure of PLA

Like other semicrystalline polymers, PLA also exhibits multiple crystalline forms. The most common crystalline form is α-form with 10_3_ helices packed in an orthorhombic unit cell and the following cell parameters: a = 1.07 nm, b = 0.645 nm, c = 2.78 nm, *α* = *β* = *γ* = 90°, *ρ*_theoretical_ = 1.247 g/cm^3^ (Figure 2a) [52]. Based on WAXD and IR results, Zhang et al. [53] first reported less-stable α’-form crystals for PLA crystallized below 120 °C. The chain conformation and crystal system of the α’ (or *δ*)-form is similar to those of α-form, the ordered chain packing is looser and less, and the cell parameters are a = 1.05 nm, b = 0.61 nm, c = 2.88 nm, *α* = *β* = *γ* = 90°, *ρ*_theoretical_ = 1.297 g/cm^3^ (Figure 2b). More studies indicate that only the α’-form crystal is mainly formed at a crystallization temperature below 100 °C, and a crystallization temperature above 120 °C forms the α-form crystal, while mixtures can be obtained between 100 and 120 °C (Figure 2c). Another form, the *β*-form crystal, was first observed by Eling et al. [54] in hot drawing of PLA melt or solution-spun fibers; the cell parameters are a = 1.031 nm, b = 1.821 nm, c = 0.900 nm, *α* = *β* = *γ* = 90 °, *ρ*_theoretical_ = 1.275 g/cm^3^. The melting temperature of the *β*-form crystal is approximately 10 °C lower than that of the α-form crystal, which indicates that the *β*-form crystal has less thermal stability [55]. Puiggali et al. [56] reported that the *β*-form crystal is a frustrated structure with three 3_1_ helices which are randomly oriented upwards and downwards in a trigonal unit cell, a = b = 1.052 nm, c = 0.880 nm, *α* = *β* = 90°, *γ* = 120°, *ρ*_theoretical_ = 1.277 g/cm^3^. The *γ*-form crystal is formed by epitaxial crystallization of PLA on hexamethylbenzene, and two chains are oriented antiparallel in the crystal cell; the cell parameters are a = 0.995 nm, b = 0.625 nm, c = 0.88 nm, *α* = *β* = *γ* = 90°, *ρ*_theoretical_ = 1.312 g/cm^3^ [57]. In addition, in the homo-crystallization of PLLA and PDLA, PLLA and PDLA can co-crystallize together into a new crystal form, a “stereocomplex”. So far, two stereocomplex crystal structures have been proposed: one is a = b = 0.916 nm, c = 0.870 nm, *α* = *β* = 109.2°, *γ* = 109.8°, *ρ*_theoretical_ = 1.274 g/cm^3^, and the other one is a = b = 1.498 nm, c = 0.870 nm, *α* = *β* = 90°, *γ* = 120°, *ρ*_theoretical_ = 1.274 g/cm^3^ [58]. Interestingly, the melting temperature of the PLA stereocomplex is approximately 50 °C higher than that of the homocrystal.

## 3. Three/Four-Dimensional Printing Technologies

Emerging as a multi-purpose material manufacturing platform, 3D printing technology is subdivided into seven categories: material jetting, binder jetting, vat polymerization, material extrusion, powder bed fusion, energy deposition and sheet lamination [61,62,63]. Four-dimensional printing technology is able to magnify the characteristics of stimuli-responsive materials [64]. Herein, we discuss recent progress in elucidating the 3D/4D printing technologies (Figure 3). The advantages and disadvantages of 3D/4D printing technologies are summarized in Table 1.

**Figure 3 ijms-24-13691-f003:**
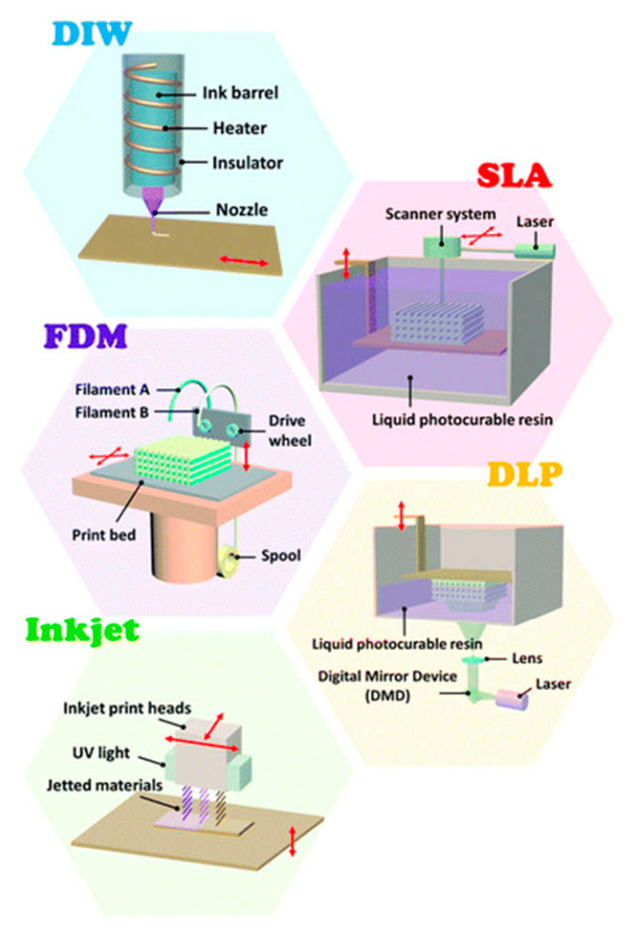
A schematic illustration of 3D/4D printing technologies (Reprinted with permission [65]. Copyright©2022, The Royal Society of Chemistry).

**Table 1 ijms-24-13691-t001:** Comparison of 3D/4D printing technologies.

Categories	Advantages	Disadvantages	Ref.
Direct ink printing	Fast printing speed Cost-effective	Low resolution (50–1000 µm) Low printing speed	[66,67]
Fused deposition modeling	Excellent recyclability Cost-effective	Low resolution (~100 µm) Poor surface quality and warpage	[68,69,70,71]
Stereolithography	High resolution (~100 nm) Sub-millisecond-level response time	High maintenance cost Low printing speed Poor fatigue properties	[72]
Digital light processing	High resolution (~10 µm) High printing speed High fidelity	High maintenance cost Materials limitation	[73,74]
Inkjet printing	High resolution (10–50 µm) Cost-effective Multi-ink printing	Slow printing speed Poor surface quality	[75,76]

### 3.1. Direct Ink Writing

Direct ink writing (DIW) is based on extruding ink from a pressurized syringe to create 3D objects. The syringe head is filled with printing ink which can be moved in three dimensions, while the platform remains stationary. Then, the 3D objects are created layer by layer. The major printing mechanism for DIW is that the viscosity of ink can be dramatically increased and the ink is in a gel state to maintain its shape once it leaves the nozzles [77].

### 3.2. Fused Deposition Modeling

Fused deposition modeling (FDM) is also called fused filament fabrication. Like DIW, FDM also creates 3D structures by connecting extruded filaments line by line and then layer by layer. In FDM, the solid filaments are melted into a semi-liquid state within the syringe and extruded on a fixed platform, then solidifying into the target objects. However, the DIW deposits viscous ink on the substrate and cures under UV light. In addition to the basic thermoplasticity, the printed materials must be rheologically and mechanically balanced to avoid the buckling of the feeding filament. PLA, polycaprolactone (PCL), and other thermoplastic elastomers can be printed with FDW [78].

### 3.3. Stereolithography

Stereolithography (SLA), also known as vat photopolymerization, is the earliest AM technology that uses a laser to selectively polymerize liquid photocurable resin in a vat by light-activated polymerization layer by layer to create the desired objects. Submicron scales are usually fabricated by SLA techniques instead of extrusion methods. Analogous to SLA, two-photon polymerization (TPP) is a method of curing photo-initiated resin utilizing a laser pulse. Owing to the special optical process of two-photon polymerization, the highest resolution is around 200 nm for TPP [79].

### 3.4. Direct Light Processing

Direct light processing (DLP) has been an attractive photopolymerization process due to its high resolution and cost-effectiveness. A digital screen is employed for the simultaneous curing of resin layers with square pixels inside a single image [80]. DLP is employed to manufacture a single layer of an object by spatially controlled solidification under laser. With this approach, a layer can be constructed rapidly. In addition, the properties of the constructed object can be mediated by changing the formula of the photocurable resin [81].

### 3.5. Inkjet Printing

Inkjet printing is a well-known AM process in which the droplets of printing materials are selectively deposited to create the desired objects [82]. The printers for inkjet printing usually include three three-axis motion platforms, a UV light, and inkjet printing heads. Low-viscosity fluids are sprayed from the inkjet printing heads, and as the platform moves, the ejected droplets are accurately deposited on the printing platform for curing as a solid.

In addition, selective laser sintering (SLS) is also a common printing technology. SLS is also based on powder processing. Instead of using a liquid binder, a laser beam with a controlled path scans the powders to sinter them by heating. Under a high-power laser, adjacent powders are fused together through molecular diffusion, and then the next layer of processing begins. This layer should be the removal of unbound powder to obtain the final product. The feature resolution (~100 µm) is determined by powder particle size, laser power, scanning spacing and scanning speed. SLS has some advantages: good accuracy, design complexity and so on. However, it also has limitations, including a rough surface finish and a lengthy printing time [83].

## 4. Crystallization-Controlled 3D/4D Printed PLA Properties

Crystal modification provides an inherent structural basis for the properties of polymer materials [84,85]. For example, compared to the α-form crystal, the α’-form crystal of PLA with disordered chain packing results in a lower modulus and barrier property. In this section, the correlation between the crystallization and properties of PLA material is introduced.

### 4.1. Mechanical Properties

Crystallization is strongly linked to the mechanical properties of polymer materials. Since the crystalline phase is more rigid than the amorphous phase in polymers, the modulus increases with the crystallinity. The modulus of a polymer can be defined using the Tsai-Halpin equation [86]:(6)$$G=\frac{G_{a}\left[G_{c}+\xi \left(\left(1-v_{c}\right)G_{a}+v_{c}G_{c}\right)\right]}{\left(1-v_{c}\right)G_{c}+v_{c}G_{a}+\xi G_{a}}$$
where *G* is the modulus of the sample; *G*_a_ and *G*_c_ represent the modulus of the amorphous phase and crystalline phase, respectively; *v*_c_ is the crystallinity of the volume; and *ξ* is the aspect ratio.

In addition, amorphous-state PLA exhibits the lowest thermal resistance with a heat deflection temperature (HDT) of approximately 50 °C [87]. Tang et al. [88] observed that the HDT of PLA can be increased with the crystallinity. As the crystallinity of PLA rises from 0.325 to 0.45, the Young’s modulus increases from 3250 to 4000 MPa, and the elongation at break decreases from 2.8 to 1.8% [89].

### 4.2. Barrier Property

The barrier property is crucial for polymer materials in packaging applications. It can be described by permeability (*P*) [90]:(7)P=DS

Here, *D* is the diffusivity, and *S* is the solubility.

The formation of crystalline structures can reduce the quantity of the amorphous phase that is available for the sorption of gas molecules. Therefore, both *D* and *S* decrease as the crystallinity increases. Tsuji et al. [91] reported that the H_2_O transmission rate of PLA monotonically decreased with the increase in crystallinity from 0 to 20% and reached a stable level when the crystallinity exceeded 30%. The permeability to water vapor also rapidly decreases from 1.5 to 0.9 [kg/(m^2^smPa)] × 10^14^ as the crystallinity of PLA increases from 0.325 to 0.45 [89].

## 5. Practical Applications

With 3D/4D printing technologies, it has become possible to manufacture nano/microstructures [92,93,94]. In 3D/4D printing, PLA is a competitive material due to its ideal properties, such as low *T*_g_ and great shape memory [95]. In addition, PLA can be also applied to novel applications because it presents good biodegradability and biocompatibility. We herein summarize the recent progress in 3D/4D printed PLA nano/microstructures for their emerging applications, with an emphasis on bio-inspired structures, flexible electronics and biomedical engineering.

### 5.1. Bio-Inspired Structures

Three/four-dimensional printed PLA nano/microstructures have the tremendous potential to be applied for bio-inspired structures owing to their ability to fabricate complicated structural architectures and possess high resolution [96,97]. Three/four-dimensional printing of various materials provides a chance to combine materials with multiple mechanical properties in a highly programmed manner [98]. Wei et al. [99] printed a complex self-supported 3D octopus and freeform 3D spiral with Ag-coated carbon nanofiber (Ag@CNFs)/PLA, and the hybrid Ag@CNFs were uniformly distributed in the PLA matrix, many circular cross sections (red circle) of the Ag@CNFs can be clearly observed on the fracture surface, indicating that most Ag@CNFs are oriented along the length of the filament, as shown by the scanning electron microscopy (SEM) images presented in Figure 4a. Recently, a printed integrated sensor–actuator (PISA) was created using an FDM printer with carbon black and PLA, and PISA samples were used to simulate the active finger touch to describe the potential application in humanoid robots (Figure 4b) [100]. Meanwhile, based on the resistance curve, the whole touching process was divided into three stages: the heating and shape change stage, the touching stage and the cooling stage.

In addition, silver nanowires (Ag-NWs) with PLA were employed to fabricate an electrically driven 4D printed structure as a gripper for grabbing a ball from a cold environment (Figure 4c) [101]. Meanwhile, Ma et al. [102] produced a 4D printed chrysanthemum with PLA matrix and PCL to simulate the sequential opening of a petal, and some air gaps were formed around the PCL microparticles, as shown in SEM images (Figure 4d), demonstrating the separation of the PLA matrix and PCL microparticles. As the PCL content increased from PCL_0.1_ to PCL_0.5_, the concentration of PCL microparticles increased while the particle size remained constant, which indicated that PCL microparticles were physically blended in the PLA matrix without any chemical reactions. In addition, mechanical metamaterials typically presented a narrow deformation domain, poor adaptability and a weak coupling effect between tension and torsion. To address these critical problems, Xin et al. [103] created 4D printed PLA-based pixel mechanical metamaterials with excellent programmability, reconfigurability and adjustability which can offer good protection for an egg falling freely from a height, as illustrated in Figure 4e. The distribution of ligaments could be classified into the left-handed chiral model and the right-handed chiral model, and the torsion of mechanical pixels could be regulated by changing the chiral model.

### 5.2. Flexible Electronics

The advancement of flexible electronic technology has improved human–machine interactions in widespread applications [104,105]. Three/four-dimensional printing in the manufacturing of PLA-based flexible electronics has aroused much attention [106]. Meanwhile, crystallization is the key to achieving high-performance materials. Smith et al. [107] developed PLA nanotubes as a “soft” piezoelectric interface onto which human skin fibroblasts can easily attach, and the level of attachment can be adjusted by controlling the crystallinity of PLA nanotubes (Figure 5a). Chizari et al. [108] reported a preparation of highly conductive carbon nanotube (CNT)/PLA nanocomposites for the functional optimization of the liquid sensors (Figure 5b). When a liquid sensor is immersed in a liquid, the relative resistance change (RRC) can be rapidly increased due to the polymer expansion stemming from the diffusion of the liquid inside the polymer matrix. When the sensor is removed from the liquid, the RRC gradually decreases. In addition, the rapid prototyping capability of 3D/4D printing provides a convenience for exploring the structural optimization of flexible electronics. Wan et al. [109] designed a conductive poly (d,l-lactide-co-trimethylene carbonate) (PLMC)/CNT ink to print shape-changing liquid sensors (Figure 5c). When voltage is applied to the printed sensors, their shape will change due to the electrothermal effect. This shape-changing capability enhances the environmental adaptability of printed sensors and will not affect the sensitivity of sensors.

In addition, Zeng et al. [110] fabricated electro-induced continuous carbon-fiber-reinforced shape memory poly (lactic acid)-based composites (CFRSMPC) with excellent bending resistance using 4D printing technology (Figure 5d). Carbon fiber bundles acted as the resistance heating elements to generate Joule heat at a specific voltage, providing a thermal stimulus for the shape recovery process of CFRSMPC. When the sample was bent from its initial shape to a temporary *V*-shape, the interlayer spacing of graphite molecules increased (*H*_initial_ < *H*_temporary_), and the density of π electrons that could move freely between the layers decreased, leading to an increase in material resistivity. As the sample was returned from a temporary shape to a permanent shape, the interlayer spacing of graphite molecules decreased (*H*_temporary_ > *H*_permanent_), and the density of state of π electrons increased, resulting in a decrease in resistivity. In addition, when an external load is applied, the carbon–carbon covalent bond in the graphite layer may break, which will also lead to a reduction in the electrical conductivity of the material. To enhance the electrical conductivity of 3D printed objects, Shi et al. [111] adopted a local enrichment strategy to fabricate PLA composite objects enriched with carbon nanotubes using 3D printing technology (Figure 5e). When the LES-PLA/CNT-part FDM waveform part (local enrichment strategy) was connected to both ends of the circuit, the light was on. Through this strategy, the electrical conductivity of 3D printed objects was dramatically enhanced, about eight orders of magnitude higher than that of traditional 3D printed objects.

### 5.3. Biomedical Engineering

Biomedical engineering is an interdisciplinary field that utilizes engineering principles and methodologies to solve problems in biology and medicine [112,113,114]. Three/four-dimensional printing provides a powerful tool to achieve the required conceptual and structural design. Interventional embolization has great potential in the treatment of aneurysms because of its safety and minimal trauma. The role of an embolization coil is to separate the aneurysm sac from the blood circulation, offering an embolic agent to the sac and reducing the risk of rupture [115]. Lin et al. [116] designed a 4D printed polybutylene succinate (PBS)/PLA-based composite vascular model that had outstanding shape memory performance and presented a broad application prospect in the biomedical engineering field, as shown in Figure 6a. The partly slight surface structure was caused by layer deposition during the printing, as shown in SEM images. The obvious mismatch between the mechanical properties of an implanted medical device and those of biological tissue is liable to cause wear, perforation and some serious complications. Lin et al. [117] also developed a 4D printed patient-specific absorbable left atrial appendage occlude (LAAO) that can match the deformation of left atrial appendage tissue to decrease complications. Meanwhile, compared to the undegraded samples, the fracture surface morphologies of degraded samples became smoother, and the reduction in cracks meant a decrease in toughness (Figure 6b).

In addition, 3D/4D printing technologies allow for the manufacturing of customizable products, such as stents [118,119]. Wang et al. [120] designed a 4D printed PLA shape memory stent with a high recovery ratio and good mechanical properties. Only a small number of blood cells could be observed on the surface of stents, meaning that the printed stents exhibited good hemocompatibility. These 4D printed stents display less medical risk and serve as intervention devices for vascular diseases (Figure 6c). Furthermore, 4D printing technology can also be used for abnormal treatment in the field of orthopedics. Wang et al. [121] prepared black phosphorus (BP) nanosheets and osteogenic-peptide-reinforced *β*-tricalcium phosphate (TCP)/P(DLLA-TMC)-based bone scaffolds using 4D printing technology, which showed a hierarchically porous structure from the SEM images, as illustrated in Figure 6d. The printed scaffolds present excellent mechanical properties comparable to those of human cancellous bone.

**Figure 6 ijms-24-13691-f006:**
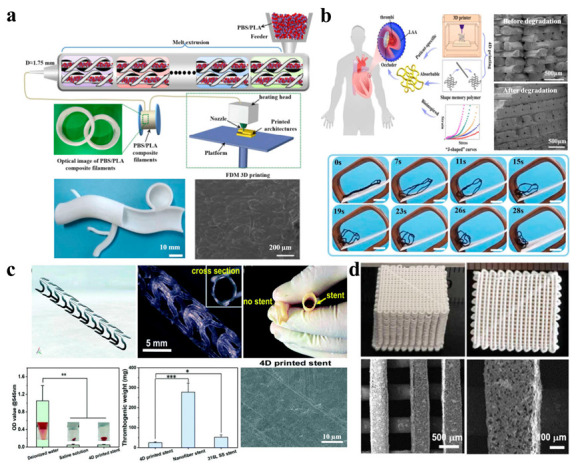
(**a**) The process flow diagram of PBS/PLA composite filaments using FDM 3D printing technology and the 4D printed PBS/PLA-based composite vascular model (Reprinted with permission [116]. Copyright©2021, Elsevier Ltd.). (**b**) Schematic of LAA tissue and 4D printing of bioinspired absorbable LAAO (the stress–strain curves of four printed 2D networks (hex55, hex66, hex77, and hex88) and LAA tissue, from left to right) (Reprinted with permission [117]. Copyright©2021, American Chemical Society). (**c**) Four-dimensional printed stent and its properties (* *p* < 0.05, ** *p* < 0.01, *** *p* < 0.001) (Reprinted with permission [120]. Copyright©2022, The Royal Society of Chemistry). (**d**) Morphology of 4D printed BP/peptide/TCP/P(DLLA-TMC)-based nanocomposites scaffolds (Reprinted with permission [121]. Copyright©2020, IOP Publishing Ltd.).

## 6. Conclusions and Future Perspectives

Three/four-dimensional printing technologies are the most effective way to create complicated functional structures. PLA is an attractive material in the 3D/4D printing field due to its low cost, biocompatibility, biodegradability and excellent shape memory properties. Meanwhile, PLA is also a semicrystalline polymer, and crystallization is essential for its physical properties. In this review, the main progress in 3D/4D printed PLA nano/microstructures was summarized and discussed with practical applications. Some 3D/4D printing technologies, such as direct ink writing, fused deposition modeling, stereolithography, direct light processing and inkjet printing, and their advantages and disadvantages were reviewed. We presented 3D/4D printed PLA products in practical applications including bio-inspired structures, flexible electronics and biomedical engineering. Furthermore, the crystallization principle and crystalline structure–property relationship were introduced in depth. Combining a flow field and self-nucleation may provide a practical method to improve the crystallization rate of PLA.

Despite the above advantages, the research on 3D/4D printing still faces some challenges, especially in practical applications. First, to achieve the rapid processing of multiscale structures, 3D printing technology with powerful functions has aroused huge academic curiosity. But until now, the technology used for 4D printing has been very limited. In addition, as the fourth dimension provides “life” to the printed objects, it is necessary to carefully identify the operation of both the original and transformed configurations. Nevertheless, researchers spend much time and effort on the materials, but there has been little work on the design challenges for this promising manufacturing technology.

## Figures and Tables

**Figure 2 ijms-24-13691-f002:**
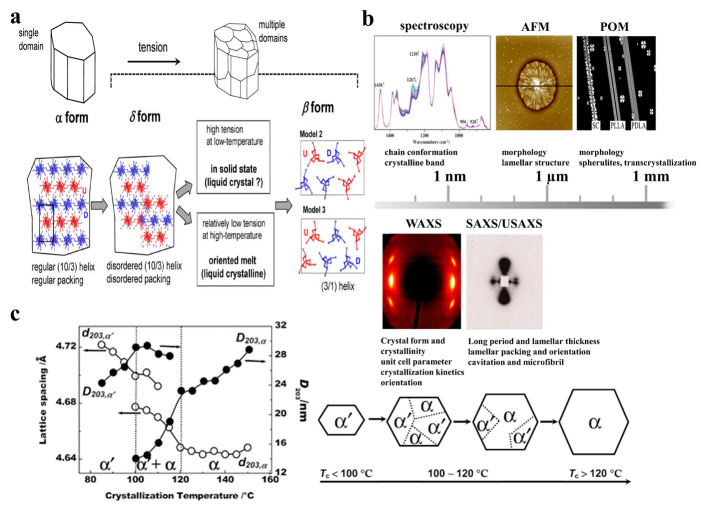
(**a**) A schematic illustration of the phase transition from *α*-form to *β*-form induced by tension (Reprinted with permission [59]. Copyright©2017, American Chemical Society). (**b**) Several typical in situ characterization techniques with different length scales (Reprinted with permission [19]. Copyright©2014, Wiley-VCH). (**c**) The crystal forms obtained under various crystallization temperatures (Reprinted with permission [60]. Copyright©2008, American Chemical Society).

**Figure 4 ijms-24-13691-f004:**
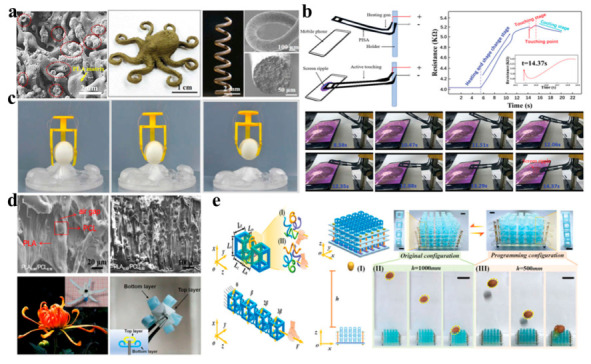
(**a**) The SEM image of a cross-section of a filament for Ag@CNF/PLA and complex 3D octopus, freeform 3D spiral. (Reprinted with permission [99]. Copyright©2019, American Chemical Society). (**b**) The practical application of actively touching a mobile phone screen (Reprinted with permission [100]. Copyright©2020, Wiley-VCH). (**c**) The 4D printed electrically controlled PLA/Ag-NW composite structure (Reprinted with permission [101]. Copyright©2020, Elsevier Ltd.). (**d**) Photograph of a 4D printed chrysanthemum with PLA/PCL composites (Reprinted with permission [102]. Copyright©2021, Springer). (**e**) Four-dimensional printed PLA-based pixel mechanical metamaterials, (I) CAD model. (II) The egg falls freely onto the original configuration (h = 1000 mm). (III) The egg falls freely onto the programming configuration (h = 500 mm) (Reprinted with permission [103]. Copyright©2021, Wiley-VCH).

**Figure 5 ijms-24-13691-f005:**
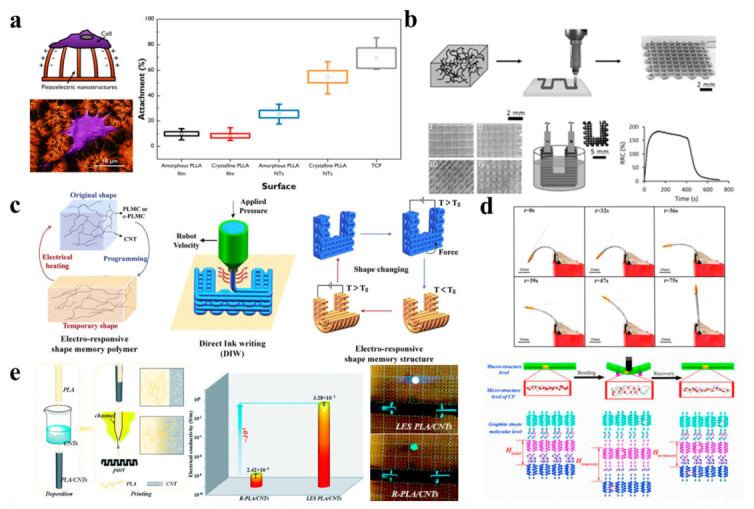
(**a**) The relationship between cell attachment and crystallinity for PLLA nanotubes as flexible piezoelectric structures (Reprinted with permission [107]. Copyright©2020, American Chemical Society). (**b**) The preparation and liquid sensitivity testing of CNT/PLA structures shown in schematics and SEM images (Reprinted with permission [108]. Copyright©2016, Wiley-VCH). (**c**) Three/four-dimensional printing of shape memory nanocomposites (Reprinted with permission [109]. Copyright©2019, Elsevier Ltd.). (**d**) Shape recovery of 4D printed CFRSMPC induced by Joule heating and the phenomenological model from three structural levels (macro-structure level, micro-structure level of CF and graphite sheets molecular level) (Reprinted with permission [110]. Copyright©2020, Elsevier Ltd.). (**e**) The electrical conductivity of the traditional R-PLA/CNT part (reference sample) and LES-PLA/CNT part (Reprinted with permission [111]. Copyright©2019, The Royal Society of Chemistry).

## Data Availability

Not applicable.

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
