# Peer review of "Three/Four-Dimensional Printed PLA Nano/Microstructures: Crystallization Principles and Practical Applications"

_ijms, 2023, doi:10.3390/ijms241813691_

Round 1

Reviewer 1 Report

The authors have crafted a comprehensive review paper on the 3D/4D printing of nano/microstructured materials, delving into their emerging applications. The paper seamlessly introduces the underlying principles of chemistry, mechanics, and rheology in a good workflow. With minor grammatical adjustments throughout the manuscript, I believe it will be well-suited for acceptance.

Some grammatical errors were detected such as;

- In line 258, provides “a” chance to combine materials with multiple mechanical properties in a highly

- In line 263-264, tuator (PISA) was created by “a” FDM printer with carbon black and PLA, and using PISA “samples” to simulate the active finger touch to describe the potential application in human- 264

I recommend a thorough review of the entire manuscript to address these and any other potential errors.

Reviewer 2 Report

This review summarizes and discusses the 3D/4D printed PLA nano/micro-structures that have been mainly published in the recent five years, and the relationship of fabrication processes and crystallization principles. In the introduction of the manuscript, only basic information about PLA nanomaterials is given.

I have the following comments on the manuscript:

*/ line 30-31: The difference between 3D and 4D printed is insufficiently described. General information is provided only. Since this is a type of manuscript - review - it is desirable that the details of 4D printing be given here.

*/ line 47: It is advisable to add basic information about the use of PLA nano/micro-structures. What is the experience with 3D printing of the PLA nanostructures. Detail information about crystalline forms of PLA (alfa, beta,  delta, ...).

*/ line 76: align the equation.

*/ line 218: the authors state that the conditions of 3D/4D printing are dependent on the mechanical properties and Barrier Property of PLA. It provides only theoretical information. For completeness, it is advisable to add the results and comparison of real measurements.

*/ page 8, line 229 + line 239: not numbered equation, different font format.

*/ In chapter 5, 3 major practical examples of 3D/4D PLA printing are presented, which can be used for their biodegradable and biocompatible properties.

Reviewer 3 Report

In general, I liked the article, it corresponds to the title and abstract. Review suggestions (if cited, give me a Hirsch 4):

Chapter 2: The text details the principles of crystallization. Nevertheless, from my point of view, this chapter can include practical recommendations for the study of polymer crystallization: in particular, about kinetic crystallization (https://doi.org/10.1038/s41428-021-00581-0), crystallization half-periods ( https://doi.org/10.32362/2410-6593-2022-17-2-164-171). I would like to see more papers on polymer crystallization published in recent years.

Chapter 3: Some methods are not covered, most notably the common Selective Laser Sintering (SLS) method. Line 188: Is it possible to use the term "semi-liquid state" in this context?
